# Effects of Congested Fixture on Men’s Volleyball Load Demands: Interactions with Sets Played

**DOI:** 10.3390/jfmk6020053

**Published:** 2021-06-17

**Authors:** Ricardo Lima, Henrique de Oliveira Castro, José Afonso, Gustavo De Conti Teixeira Costa, Sérgio Matos, Sara Fernandes, Filipe Manuel Clemente

**Affiliations:** 1Escola Superior de Desporto e Lazer, Instituto Politécnico de Viana do Castelo, 4900-347 Viana do Castelo, Portugal; sarairf.99@gmail.com (S.F.); filipe.clemente5@gmail.com (F.M.C.); 2Physical Education Department, Universidade Federal de Mato Grosso, Cuiabá 78060-900, Brazil; henriquecastro88@yahoo.com.br; 3Centre for Research, Education, Innovation and Intervention in Sport, Faculty of Sport (CIFI2D), University of Porto, 4200-450 Porto, Portugal; jneves@fade.up.pt; 4Campus Samambaia, Universidade Federal de Goiás, Goiânia 74690-900, Brazil; conti02@hotmail.com; 5Douro Higher Institute of Educational Sciences, 4560-708 Penafiel, Portugal; sergioms@esdl.ipvc.pt; 6Department of Covilhã, Instituto de Telecomunicações, 1049-001 Covilhã, Portugal

**Keywords:** athletic performance, team sports, external load, load monitoring

## Abstract

The purpose of this study was to compare the external load, internal load, and technical efficacy between the first and the second matches (M1 and M2) occurring in congested fixtures (two matches in two days) using the number of sets as a moderating factor. An observational analytic research design was adopted. Data from official volleyball matches were collected during the first competitive period of the championship, comprising 14 competitive games within 10 weeks. Ten male elite volleyball athletes (age: 21.7 ± 4.19 years of age; experience: 6.2 ± 3.8 years; body mass: 85.7 ± 8.69 kg; height: 192.4 ± 6.25 cm; BMI: 23.1 ± 1.40 kg/m2) participated in this study. Players were monitored for external load (number of jumps and height of jumps) and internal load (using the rate of perceived exertion—RPE). Additionally, notational analysis collected information about attack efficacy and receptions made during matches. The mixed ANOVA revealed no significant interaction between time (M1 vs. M2) and number of sets for number of jumps per minute (*p* = 0.235; $\eta_{p}^{2}$ = 0.114), mean jump height (*p* = 0.076; $\eta_{p}^{2}$ = 0.193), RPE (*p* = 0.261; $\eta_{p}^{2}$ = 0.106), attack efficacy (*p* = 0.346; $\eta_{p}^{2}$ = 0.085), Positive reception (*p* = 0.980; $\eta_{p}^{2}$ = 0.002) and Perfect reception (*p* = 0.762; $\eta_{p}^{2}$ = 0.022). In conclusion, congested fixtures do not seem to affect the performance of volleyball players negatively.

## 1. Introduction

Monitoring training load in sports is essential for planning training sessions and preparing for matches [1,2]. Training load has been investigated to assess both internal and external loads [3,4]. Internal load is characterized by the physiological and psychological stress imposed on athletes during training sessions and/or competitions, such as training impulse (heart-rate based method), session rating perceived exertion—sRPE (multiplying session duration with rating perceived exertion—RPE, monotony (daily mean of weekly training load), and strain (total weekly training load multiplied by monotony) [5], while external load is the workload exerted by an athlete regardless of internal physiological and psychological changes [6]. In this sense, the methods for monitoring training load must be adjusted according to the specificity and demands of each sport to provide athletes and coaches with important information [6,7,8].

As a modality in which the possession of the ball changes cyclically during the rally [9], volleyball represents a unique balance in demand between the tactical, technical, physiological, and physical dimensions of the game [10,11]. Volleyball is characterized by intermittent efforts of short duration and high-intensity, interspersed with brief rest periods [12]. Furthermore, in this sport, several physical attributes and skills are required, such as lower-limb explosive power (combinations of repetitive jumps) and agility in displacement to cover short distances (multidirectional movements), and these are performed over long and extended matches [13,14,15].

The success of the training process and, consequently, of volleyball matches depends on adjustments that balance the magnitude and distribution of training load and recovery [12,16]. Positive relationships were observed between performance in game actions that require powerful vertical jumps (serving, blocking, and attacking) and overall success in matches [14]. Thus, vertical jump height has been considered as a good performance parameter when analyzing athletes in this sport when considered among other technical actions, such as setting, attacking, blocking, and serving—thus, jumping is a very important task during volleyball training and matches [3,13,17,18].

Vertical jump performance and internal training load have been widely used to quantify and monitor loads imposed on athletes during volleyball training and matches [3,4,12,18,19,20,21,22]. Thus, monitoring vertical jumping and training load in this sport is essential to understand the demands of training and matches and, consequently, the adaptation process inherent to workload [23,24,25]. This fact is in line with the recent historical perspective reflected by Foster et al. [26], which confirmed that the training impulse concept and internal workload assessed by the RPE are able to account for both positive and negative training outcomes and could be tools to understand the training process as a way to optimize outcomes. Consequently, the competitive calendar in team sports requires an optimization of the training plan and organization, mainly due to the short preparation time and the need to maintain athletes’ high levels of performance in competitions [27,28].

Additionally, it is known that accumulating several matches over a short period negatively affects players’ performance and well-being while increasing the rate of injury because players are given insufficient time for recovery [21,29,30,31]. Thus, coaches should manipulate the training workload of the different daily and weekly sessions of training to prepare athletes for competitions [21] optimally. Most research on the effects of congested fixtures has been done in soccer [30,32,33,34,35,36], and few studies approached this subject in volleyball [21,37,38,39]. However, when compared to soccer, basketball, and handball, volleyball is characterized by smaller increases in inflammation (240% in soccer, 120% in handball and basketball, and 90% in volleyball) and muscle damage markers (115% in soccer, 110% in basketball and handball, and 80% in volleyball) [37].

Moreover, regarding psychophysiological parameters in elite male volleyball players in the context of congested fixtures, increases in post-match cortisol values were observed when baseline values were compared to post-match values, but no differences were observed between matches [38].

For the aforementioned, the purpose of this study was to compare the external load (standardized number of jumps per match of all the team, the mean jump height), internal load (RPE), and technical efficacy between the first and the second matches occurring in congested fixtures (two matches in two days) using the number of sets as a moderating factor.

## 2. Materials and Methods

### 2.1. Participants

Ten male elite volleyball athletes (two setters, three middle blockers, three outside hitters, and two opposites) of an intermediate rating team from the Portuguese 1st Division (age: 21.7 ± 4.19 years of age; experience: 6.2 ± 3.8 years; body mass: 85.7 ± 8.69 kg; height: 192.4 ± 6.25 cm; BMI: 23.1 ± 1.40 kg/m^2^) who regularly practice seven times per week (105 ± 12.4 min), participated in this study. The players were monitored in official matches over the course of 10 weeks, during the first phase of the Portuguese Championship (October 2020 to December 2020). The following inclusion criteria were used: (i) players did not have injuries or illnesses during the period of data collection, and (ii) players participated in both matches of each congested weekend during the period of analysis. All the players voluntarily participated in the study and were informed about the study’s design, implications, risks, and benefits. After receiving this information regarding the study, the players signed an informed consent. They were free to withdraw from the research if they so wished. The study was conducted in line with the international ethical guidelines for sport and exercise science research recommended by the Declaration of Helsinki [40]. In addition, the study protocol was approved by a local university ethics committee before the data was collected (CTC-ESDL-CE003-2020).

### 2.2. Experimental Approach to the Problem

An observational research design was adopted. Data from official volleyball matches were collected during the first competitive period of the championship (October 2020 to December 2020), comprising 14 competitive games within 10 weeks. The matches in weeks 3, 5, and 10 were congested, with two matches played during each weekend (Table 1). For the purposes of our analysis, only athletes who played in both matches in a congested week were included in the study. A comparative research design tested the differences between internal and external load measures, as well as technical efficacy. Players were monitored for external load in all matches using an inertial measurement unit (IMU) and internal load answering to the RPE [41,42].

### 2.3. Training Load Monitoring

#### 2.3.1. Internal Load

Familiarization with Borgs’ scale (CR10) [42] was previously presented to the team, tested for two weeks, and applied daily across the three weeks of the pre-competitive season to minimize error and increase the accuracy of the answers. Approximately 30 min after the end of training sessions or matches, players were asked, “How was your workout?” Internal load (rating of perceived exertion-based training load (sRPE)), reported as arbitrary units (A.U.), was calculated by multiplying the adapted version of Borg’s scale. The sRPE has been applied in several studies and is considered a valid surrogate of internal load [4,13,26,43,44,45].

#### 2.3.2. External Load

An IMU consisting of 3-axis gyroscopes, 3-axis magnetometers, and 3-axis accelerometers (Vert^®^ Classic, MyVert, Florida, FL, USA) was used by each player during the data-collection period to determine the external load of each match. This unit estimated the number of jumps performed and the height of each jump. The inertial sensor was placed on a belt secured to athletes’ hips before warming up, but the data were collected only after the beginning of each match. This instrument is a valid tool for field-based jump load measurements and has been used in other studies [1,13,46,47]. The data were transmitted via Bluetooth to an application (MyVert Coach from IOS), making it possible to monitor all athletes in real-time. The device presents a mean error of −2.4 cm when compared to gold-standard methods, such as force plate or video system [47]. The same IMU was always used for the same player to avoid variability between devices.

### 2.4. Technical Efficacy

After recording the matches with a SONY FDR-AX33 (XAVCS 4K: 3840 × 2160) positioned behind the court (see Figure 1), technical actions were analyzed using Data Project—Data Volley, a statistical software program that is often used to analyze technical and tactical actions in volleyball because it analyzes the game from a holistic view, considering data, such as individual skill actions and the general stats of teams, with video analysis [48].

The technical variables included in this study were: (a) receptions (positive B/C; and perfect—A) and (b) attacks. The efficacy of the reception was established in relation to setting conditions, as reported in a previous study [49]. A positive reception was considered when the setter had only two hitters available to set the attack, while a perfect reception was considered when the setter had all options available for attacking. Attack efficacy was calculated by subtracting the sum of attack errors and attacks blocked from the number of attack points [3].

### 2.5. Statistical Procedures

All match analyses were done by a single observer with 15 years of experience in elite volleyball in Portugal, but Coleman [50] suggested these software packages allow users to achieve intraobserver reliability values of between 0.96 and 1.00. In the case of this study, 33% of the full data set (2 matches) was used to test intraobserver reliability. An intraclass correlation coefficient (ICC) test was executed to analyze the reliability levels (poor, below 0.50; moderate, 0.50 to 0.75, good, 0.76 to 0.90; excellent, 0.91 to 1.00) (Post, 2016). The ICC revealed excellent intra-reliability (ICC = 0.91) values (Table 2).

Normality and homogeneity of the sample were first tested and confirmed, using the Kolmogorov–Smirnov test (*p* > 0.05), as well as Levene’s test (*p* > 0.05). A mixed ANOVA (time*sets) tested the variation of matches 1 and 2 performed during the same weekend using the number of sets as a factor. Mauchly’s test presented no sphericity; thus, the Greenhouse–Geisser correction was used for correcting the violation of sphericity. Since no significant interactions were found between time*factor, the repeated measures ANOVA was used to determine variations between matches 1 and 2. The effects size (ES) for repeated measures ANOVA was calculated using the partial eta squared ($\eta_{p}^{2}$). The magnitude of $\eta_{p}^{2}$ was made following these thresholds [51]: < 0.04, trivial effect size; 0.04–0.25, minimum effect size; 0.25–0.64, moderate effect size; >0.64, strong effect size. All the statistical procedures were executed in the SPSS Statistics (version 25.0; IBM, Armonk, NY, USA) for a *p* ≤ 0.05.

## 3. Results

Descriptive statistics of external load and technical execution in matches 1 and 2 organized by the number of sets per match can be observed in Table 3.

The mixed ANOVA revealed no significant interaction between time (M1 vs. M2) and number of sets for number of jumps per minute (*p* = 0.235; $\eta_{p}^{2}$ = 0.114), mean jump height (*p* = 0.076; $\eta_{p}^{2}$ = 0.193), RPE (*p* = 0.261; $\eta_{p}^{2}$ = 0.106), attack efficacy (*p* = 0.346; $\eta_{p}^{2}$ = 0.085), POS reception (*p* = 0.980; $\eta_{p}^{2}$ = 0.002), and PERF reception (*p* = 0.762; $\eta_{p}^{2}$ = 0.022).

Repeated measures ANOVA did not reveal significant changes between M1 and M2 for number of jumps per minute (*p* = 0.229), mean jump height (*p* = 0.451), RPE (*p* = 0.472), attack efficacy (*p* = 0.671), POS reception (*p* = 0.771), or PERF reception (*p* = 0.433). Figure 2 presents the percentage of change (M2-M1) for the different measures. Additionally, no differences were found between sets (*p* > 0.05) for all the measures.

## 4. Discussion

Our goal was to assess whether congested fixtures (i.e., weeks with two matches in one weekend) affected the vertical jump performance, attack efficacy, and reception quality in male 1st division volleyball players who played in both matches during such weeks. There were only three congested fixtures during the first phase of the national championships. In the first and third congested fixtures, eight players competed in both matches, while in the second fixture, ten players competed in both matches. There was no discernible tendency with regards to positional status (i.e., setter, opposite, middle-blocker, or outside hitter). Statistical analyses revealed no significant differences between the number of sets or match 1 and match 2 (within each congested fixture) for any of the variables analyzed. Therefore, in this limited sample, congested features did not interfere with the players’ performance. However, the small size of our sample suggests that it lacks statistical power [52], and so a larger sample could have provided different results.

Indeed, an analysis of descriptive data showed a decrease in the number of jumps per minute between three-set and the four-set matches, suggesting a decreased intensity of play in four-set matches—however, this finding is speculative, as our data did not enable testing of this hypothesis. Furthermore, although four-set and five-set matches may appear more demanding and balanced than three-set matches [3,53], this is not always the case. For example, a three-set match may have sets of 28–26, 27–25, and 30–28 (denoting a very balanced match). Conversely, a five-set match may have sets of 25–10, 11–25, 25–12, 10–25, 15–6. These situations are not uncommon in volleyball. So, the relationship between the number of sets and the balance within the match is not linear. Furthermore, in this example, both the three-set match and the five-set match had a total of 164 points played. Therefore, perhaps the number of sets is not a measure of match load, whereas the total number of points played could potentially provide a more relevant metric.

An inspection of descriptive results also suggested no trend in terms of mean jump height, with this variable presenting a relatively homogeneous behavior regardless of the number of sets of match order (i.e., the first versus the second match within a congested fixture), confirming previous findings [3] with a similar sample. In our data—notwithstanding a few idiosyncratic exceptions—RPE, attack efficacy, and reception quality did not depend on match order, or the number of sets played, either.

Congested fixtures have been suggested to affect performance and well-being in team sports, such as basketball [54], futsal [55], handball [38], soccer [33], and volleyball [21]. In volleyball, matches during congested weeks had significantly higher session-RPE scores than matches in regular weeks (364.71 A.U. versus 252.97 A.U., *p* < 0.05) [20]. In high-level volleyball players, it has been shown that stress tolerance is negatively affected by congested fixtures [38]. However, it is possible that congested fixtures have different effects on players of different levels of practice [55], a topic that should be further explored in the literature.

Furthermore, a relevant point that was not addressed in our study is what happens after the congested fixtures and/or if there are too many congested fixtures during a season. For example, in a study on 15 under −20 elite male soccer players exposed to 14 matches over eight weeks (±1.75 matches per week), sprint velocity, change in direction speed, and mean propulsive power were impaired after the match-congested period [34]. However, this was an observational, single-arm study, and so it is impossible to assess whether eight weeks of non-congested fixtures would have produced different effects.

Another study on 40 athletes analyzed the influence of congested fixtures on professional Brazilian soccer players over three years (59 matches) and demonstrated that high-intensity activities varied according to the player (positive to negative relationships with high-intensity), the tactical formation of the team, game status (victory or defeat), match location (home game or away game), among other aspects, indicating variability in performance based on the context [36]. Another study with 42 soccer players across three seasons compared congested and non-congested weeks (i.e., two matches versus one match per week) [35], with the results suggesting that injury rates increased by 20% during multi-match weeks. Therefore, even if congested weeks do not affect match performance, they may still have consequences for recovery and, thus, require specialized training load management [21,56,57].

## 5. Conclusions

Congested fixtures do not seem to affect the performance of volleyball players negatively. Matches 1 and 2 of each weekend during congested weeks presented similar outcomes. However, studies with a more refined performance analysis (e.g., considering playing role) should be conducted to detect relevant differences.

## Figures and Tables

**Figure 1 jfmk-06-00053-f001:**
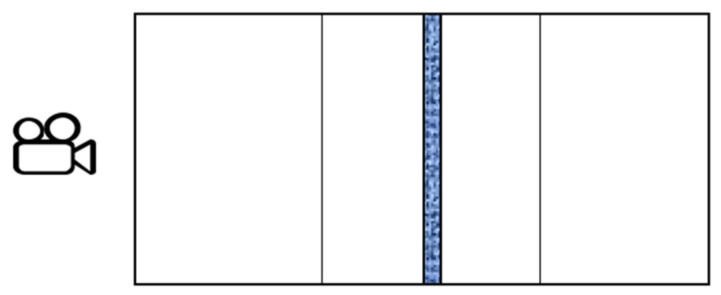
Video Cam SONY FDR-AX33 positioned behind the pitch.

**Figure 2 jfmk-06-00053-f002:**
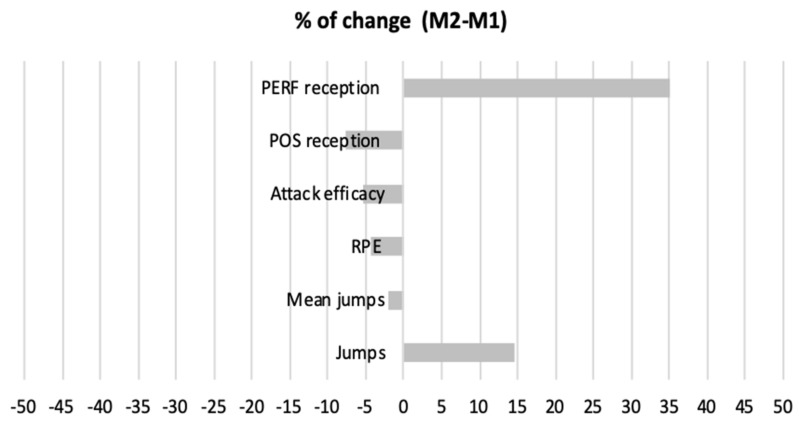
Percentage of change (M2-M1) for the different outcomes. M1—match 1; M2—match 2; POS—positive; PERF—perfect; RPE—rate of perceived exertion.

**Table 1 jfmk-06-00053-t001:** Characteristics of the congested week schedules and classification at the moment.

Weeks of the Competition Phase	Match 1	Match 2	Players Repeating the Matches (n)	Positional Status
Week 3	Sets: 3 Score: Victory Relative Classification: +7	Sets: 3 Score: Defeat * Relative Classification: −6	8	Setter—1 Opposite—2 Middle Blocker—3 Outside Hitter—2
Week 5	Sets: 4 Score: Defeat Relative Classification: −4	Sets: 4 Score: Defeat Relative Classification: −6	10	Setter—2 Opposite—2 Middle Blocker—3 Outside Hitter—3
Week 10	Sets: 5 Score: Victory Relative Classification: −2	Sets: 3 Score: Victory Relative Classification: +4	8	Setter—2 Opposite—1 Middle Blocker—2 Outside Hitter—3

* Relative Classification = classification according to the position of the team and the opponent (e.g., Team at 1st place and opponent in 3rd place = +2 positions).

**Table 2 jfmk-06-00053-t002:** Intraobserver reliability (Intraclass correlation coefficient).

Variable	Intraobserver Reliability	Interpretation
Positive Reception (POS)	0.86	Good
Perfect Reception (PERF)	0.97	Excellent
Attack Efficacy	0.91	Excellent

**Table 3 jfmk-06-00053-t003:** Descriptive statistics (mean ± standard deviation) of outcomes between the first and second match considering the number of sets involved.

	3-3 Sets		4-4 Sets		5-3 Sets	
	M1	M2	M1	M2	M1	M2
Jumps (n/min)	0.86 ± 0.30	0.84 ± 0.30	0.59 ± 0.18	0.59 ± 0.29	0.61 ± 0.22	0.95 ± 0.66
Mean jump height (cm)	57.2 ± 5.1	52.1 ± 9.9	52.5 ± 10.5	56.8 ± 12.1	52.6 ± 7.1	49.4 ± 13.5
RPE (A.U.)	6.2 ± 1.0	6.8 ± 0.7	7.1 ± 1.1	6.8 ± 1.9	8.0 ± 1.1	6.9 ± 2.0
Attack Efficacy (%)	43.0 ± 27.9	30.6 ± 25.3	31.1 ± 22.7	36.1 ± 31.7	37.4 ± 24.6	38.3 ± 28.7
POS reception (%)	23.9 ± 36.6	23.7 ± 37.8	33.4 ± 44.6	30.1 ± 32.2	50.0 ± 45.2	45.3 ± 49.9
PERF reception (%)	13.7 ± 27.3	19.6 ± 34.5	17.1 ± 32.7	16.2 ± 18.1	12.1 ± 16.8	24.0 ± 37.1

A.U.: arbitrary units; M1: match 1; M2: match 2; POS: positive; PERF: perfect; RPE: rate of perceived exertion; RPE: rate of perceived exertion.

## Data Availability

The data sharing is not applicable.

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
