# Peer review of "Effects of Congested Fixture on Men’s Volleyball Load Demands: Interactions with Sets Played"

_jfmk, 2021, doi:10.3390/jfmk6020053_

Round 1
Reviewer 1 Report
General comments
The paper aimed at comparing the external load (standardized number of jumps per match of all team, the mean jump height), internal load (rate of perceived exertion scale - RPE), and technical efficacy between the first and the second matches occurring in congested fixtures (two matches in two days) using the number of sets as a moderating factor.
The paper is generally well written based on sound literature, the methods are clear, detailed and replicable, the results well-presented and discussed with respect to the literature. However, I have some minor concerns about the introduction and statistical approach (see comments)
When you use the abbreviation once, make sure you will explain it first and then you would use it for all the manuscript in a consistent manner.
Specific comments
Introduction
The authors kept in mind that this section is a development of the hypotheses of the study leading to the purpose of the investigation. However, I would add a small section that would describe the main training load used in research and sport practice (i.e. TRIMP, session-RPE), with weaknesses and strengths. Furthermore, I would suggest the authors to check this recent review (Foster et al. 2021. 25 Years of Session Rating of Perceived Exertion: Historical Perspective and Development on the session-RPE), as the authors adopted this method but did not include the main responsible of the session-RPE method in their references.
Materials and Methods
Methods and procedures are clear, detailed and replicable.
Statistical analysis
Although it is common to use the statistical approach and design described, this method is likely to increase the probability of a Type 2 error due to your small sample size. A more powerful approach that accounts your small sample size and repeated measures design might be the repeated mixed (multilevel or hierarchical) modeling, in which you estimate different random effects or errors within and between clusters (Hopkins et al., 2009. Medicine & Science in Sports & Exercise; Lininger et al., 2015. Journal of Athletic Training).
Discussion
I think the results of the study are well discussed with respect to the current literature.
Figure and Tables
Tables and figures should stand on their own. Make sure you check this.
Reviewer 2 Report
Monitoring training load in sports is essential for planning training sessions and preparing for matches.
Author Response
Dear Reviewer. Thank you so much.
Reviewer 3 Report
Lines 78-79: I think this sentence should be incorporated in the previous paragraph, by also expanding the information (highlighting the main findings of previous studies).
Lines 124-125: RPE training load method has been indeed used in several studies and it is kind of difficult to decide which one to indicate to support this kind of statement. Therefore, my suggestion is to refer to a new published paper (Foster et al 2021, 10.1123/ijspp.2020-0599) which might be comprehensive of the whole concept.
Figure 1. Make sure the Figure can stand on its own by avoiding the use of abbreviation (or if necessary, spelled them out).
Line 272: Make sure you insert the name of the institution.
Reference 42: Please update with the final publication details (the paper had been published in 2020).
Reference 43: Please insert the full title.
Reference 45: Please update with the final publication details (the paper had been published in 2017).
Reference 55: Please update with the final publication details (the paper had been published in 2019).
